# Towards critical white ice conditions in lakes under global warming

Gesa A. Weyhenmeyer [1] ✉, Ulrike Obertegger [2], Hugo Rudebeck [1], Ellinor Jakobsson [1], Joachim Jansen [1], Galina Zdorovennova [3], Sheel Bansal [4], Benjamin D. Block [5], Cayelan C. Carey [6], Jonathan P. Doubek [7,8], Hilary Dugan [9], Oxana Erina [10], Irina Fedorova [11], Janet M. Fischer [12], Laura Grinberga [13], Hans-Peter Grossart [14,15], Külli Kangur [16], Lesley B. Knoll [17], Alo Laas [16], Fabio Lepori [18], Jacob Meier [4], Nikolai Palshin [3], Mark Peternell [19], Merja Pulkkanen [20], James A. Rusak [21,22], Sapna Sharma [23], Danielle Wain [24] & Roman Zdorovennov [3]

The quality of lake ice is of uppermost importance for ice safety and under-ice ecology, but its temporal and spatial variability is largely unknown. Here we conducted a coordinated lake ice quality sampling campaign across the Northern Hemisphere during one of the warmest winters since 1880 and show that lake ice during 2020/2021 commonly consisted of unstable white ice, at times contributing up to 100% to the total ice thickness. We observed that white ice increased over the winter season, becoming thickest and constituting the largest proportion of the ice layer towards the end of the ice cover season when fatal winter drownings occur most often and light limits the growth and reproduction of primary producers. We attribute the dominance of white ice before ice-off to air temperatures varying around the freezing point, a condition which occurs more frequently during warmer winters. Thus, under continued global warming, the prevalence of white ice is likely to substantially increase during the critical period before ice-off, for which we adjusted commonly used equations for human ice safety and light transmittance through ice.

[1]Department of Ecology and Genetics/Limnology, Uppsala University, Uppsala, Sweden. [2]Fondazione Edmund Mach, Research and Innovation Centre, San Michele all'Adige, Italy. [3]Northern Water Problems Institute, Karelian Research Centre RAS, Petrozavodsk, Russia. [4]U.S. Geological Survey, Northern Prairie Wildlife Research Center, Jamestown, ND, USA. [5]Center for Ecological Sciences, Tetra Tech, Inc., Montpelier, VT, USA. [6]Department of Biological Sciences, Virginia Tech, Blacksburg, VA, USA. [7]School of Natural Resources & Environment, Lake Superior State University, Sault Sainte Marie, MI, USA. [8]Center for Freshwater Research and Education, Lake Superior State University, Sault Sainte Marie, MI, USA. [9]Center for Limnology, University of Wisconsin-Madison, Madison, WI, USA. [10]Lomonosov Moscow State University, Moscow, Russia. [11]St Petersburg State University, St Petersburg, Russia. [12]Department of Biology, Franklin & Marshall College, Lancaster, PA, USA. [13]Laboratory of Hydrobiology, Institute of Biology, University of Latvia, Riga, Latvia. [14]Department Plankton and Microbial Ecology, Leibniz Institute for Freshwater Ecology and Inland Fisheries, Stechlin, Germany. [15]Department of Biochemistry and Biology, Potsdam University, Potsdam, Germany. [16]Centre for Limnology, Chair of Hydrobiology and Fishery, Institute of Agricultural and Environmental Sciences, Estonian University of Life- Sciences, Tartu, Estonia. [17]Itasca Biological Station, University of Minnesota, Lake Itasca, MN, USA. [18]Institute of Earth Sciences, University of Applied Sciences and Arts of Southern Switzerland, Mendrisio, Switzerland. [19]Department of Earth Sciences, University of Gothenburg, Gothenburg, Sweden. [20]Finnish Environment Institute, Jyväskylä, Finland. [21]Department of Biology, Queen's University, Kingston, ON, Canada. [22]Dorset Environmental Science Centre, Ontario Ministry of the Environment, Conservation and Parks, Dorset, ON, Canada. [23]Department of Biology, York University, Toronto, ON, Canada. [24]7 Lakes Alliance, Belgrade Lakes, ME, USA. ✉e-mail: Gesa.Weyhenmeyer@ebc.uu.se

The majority of lakes in the Northern Hemisphere are still periodically covered by ice in winter, but long-term records show a rapid decline in the number of days when lakes are frozen, mainly due to globally increasing air temperatures[1–4]. Increasing temperatures have an effect not only on the mean but also on the amplitude of the annual air temperature cycle, which in turn causes accelerated ice cover loss rates in a warmer world[5], which have been documented for many lakes across the Northern Hemisphere over the past decades[3,6]. Future climate simulations indicate that the spatial distribution of lake ice cover in the Northern Hemisphere will turn more than 35,000 seasonally frozen lakes into intermittently ice-free lakes in a 2 °C warmer world, affecting nearly 400 million people[7]. Of those lakes, up to 5700 are projected to become permanently ice-free within this century[8].

Although lake ice cover dynamics have been intensively studied, with melting and thawing mechanisms well known[9,10], lake ice quality is usually not assessed and only very few studies present data on ice quality (for examples see Table 1 and Fig. 1a). Characteristics of lake ice quality include measures on ice thickness, transparency (either clear/"black" or opaque/"white"), crystal structure, and impurities[10]. Most lake ice quality data are available for total ice thickness in part because those data can be retrieved from satellites[11]. The thickest lake ice on Earth has been observed on permanently frozen lakes that are covered by glaciers, where the present record of 295 m is from Lake Vostok in Antarctica[12]. The maximum ice thickness of seasonally frozen lakes is much less, reaching around 2 m[9]. Generally, the longer the ice cover lasts, the thicker the ice layer can become, even in small, well mixed lakes with intermittent ice cover, as reported from Müggelsee[13], a lowland lake in northeastern Germany.

The ice layer on lakes commonly consists of black and/or white ice, also referred to as congelation ice and snow-ice, respectively[14]. White ice has about half of the load-bearing strength compared to black ice[15,16]. In addition, white ice strongly reduces the penetration of photosynthetically active radiation through ice, in contrast to clear black ice which usually does not affect light penetration more than lake water[17,18]. With an increasing white ice thickness, photosynthetically active radiation underneath ice decreases and approaches zero when the white ice layer reaches a thickness of 30 cm[9]. Because light conditions control the phenology and community composition of winter

### Table 1 | Examples of lakes for which some ice quality data were available in the literature (for location of the lakes see Fig. 1a)

| Lake name | Country | Reference |
| --- | --- | --- |
| Lake Opinicon | Canada | Agbeti and Smol (1995)[49] |
| Rideau Canal | Canada | Barrette (2011)[16] |
| Upper Rock Lake | Canada | Agbeti and Smol (1995)[49] |
| Lake Kilpisjärvi | Finland | Korhonen (2006)[22] and Leppäranta (2019)[50] |
| Lake Pääjärvi | Finland | Jakkila et al. (2009)[51] |
| Lake Abashira | Japan | Ohata et al. (2016)[52] |
| Lake Haruna | Japan | Maeda and Icimura (1973)[53] |
| Atnsjøen | Norway | Jensen (2019)[54] |
| Krasne | Poland | Pasztaleniec and Lenard (2008)[39] |
| Lake Mikołajskie | Poland | Kalinowska and Grabowska (2016)[55] |
| Piaseczno | Poland | Pasztaleniec and Lenard (2008)[39] |
| Rogóźno | Poland | Pasztaleniec and Lenard (2008)[39] |
| Lake St. Ana | Romania | Felfoldi et al. (2015)[56] |
| Lake Vendyurskoe | Russia | Zdorovennova et al. (2021)[36] |
| Frains Lake | USA | Bolsenga et al. (1991)[17] |
| Great Lakes | USA | Bolsnega and Vanderploeg (1992)[35] |
| Lake Bishop | USA | Bolsenga et al. (1991)[17] |
| Lake Erie | USA | Bolsenga et al. (1991)[17] |

**a**

Permafrost in the Northern hemisphere

Durability: Continuous permafrost, Discontinuous permafrost, Sporadic permafrost, Isolated patches, Subsea permafrost

© International Permafrost Association

**b**

Winter (December-January-February)
December
January
February
March

T (°C)
Year

**c**

White ice

Black ice

and spring plankton and fish[19], knowledge about the thickness of white ice on a lake is essential for understanding lake ecology. Yet, at present, there is no global information available on the occurrence and variability of white ice on lakes. We, therefore, conducted an ice sampling campaign in lakes across the Northern Hemisphere (Fig. 1a) within the

**Fig. 1 | Sampling locations of seasonally frozen lakes and Northern Hemisphere winter air temperatures since 1880. a** Open access map from the International Permafrost Association (https://www.eea.europa.eu/legal/copyright) showing Ice-Blitz sampling locations with lake names during winter 2020/2021 (red dots). Also shown are locations of lakes for which some ice quality data from the literature are available (black dots). **b** Times series of winter and monthly mean Northern Hemisphere air temperatures (T) from 1880 to 2021, shown as anomalies over the base period 1951 to 1980. Lines are smoothing splines using a lambda of 0.05. Red dots represent air temperatures during the IceBlitz sampling campaign. Data for air temperature are from NASA GISS Surface Temperature Analysis (GISTEMP).
**c** Examples of IceBlitz sampling occasions in Estonia and Russia during white and black ice conditions, respectively (photo courtesy: Margot Sepp and Oxana Erina).

Global Lake Ecological Observatory Network (GLEON), following a standardized protocol (Supplementary Information). The campaign, named IceBlitz, was performed from December 2020 to April 2021, coinciding with one of the warmest winters in the Northern Hemisphere since 1880 (Fig. 1b). During the campaign, total ice thickness, the thickness of black and white ice (Fig. 1c) and the thickness of snow and/or slush layers on top of the ice layer were measured in 31 lakes distributed across ten countries in the Northern Hemisphere (Fig. 1a). The choice of lakes was dependent on IceBlitz participants' ability to perform winter sampling. Altogether, 167 ice quality observations were made (Fig. 1a, Table 2). For two of the 31 lakes, i.e., Mozhaysk Reservoir and Lake Vendyurskoe, long-term lake ice quality data dating back to 1971 and 1996, respectively, were available. We used these time series to evaluate how lake ice conditions in 2020/2021 deviated from ice conditions observed during previous, colder winters.

## Results and discussion
### Ice quality observations

During the IceBlitz campaign, the total lake ice thickness on lakes across the Northern Hemisphere varied from 1 to 103 cm, with a median of 29 cm (mean ± one standard deviation: 32 ± 18 cm). Most measurements were performed in January and February ($n = 53$ and 58, respectively), a few weeks after ice-on in December or January. Many lakes were already ice-free in April, decreasing the number of sampling occasions during that month to ten. In our data set, the thickest ice layers were usually observed in February or March, depending on how long the ice cover lasted.

The observed ice thickness during the 2020/2021 IceBlitz campaign was generally lower than we found reported in the literature (Table 1). Out of 117 lake ice thickness observations from 18 lakes performed in winters prior to the sampling year (Table 1, Fig. 1a), only 21 observations demonstrated an ice thickness less than 29 cm, and the median across all 117 observations from the literature was 42 cm (mean ± one standard deviation: 45 ± 16 cm). There are many possible reasons why previous observations from other lakes deviate from our observations during the very warm winter in 2020/2021 (Fig. 1b), such as geographical location, lake size, lake depth, sampling date, degree of global warming, morphological and hydrothermal conditions. It is, however, rather well documented that many lakes in the world presently experience a trend of thinner ice e.g.,[20–22], suggesting that the thinner ice conditions during our sampling campaign are a result of rapidly increasing winter air temperatures in the Northern Hemisphere during the past years (Fig. 1b). Increased winter air temperatures are most likely also the reason why we found a significantly decreased total ice thickness in the Mozhaysk Reservoir in 2011–2021 compared to 1971–1979 (median: 37 and 53 cm, respectively; Wilcoxon signed-rank test: $p < 0.0001$, $n = 210$ and 169, respectively). Because winter air temperatures in the Northern Hemisphere are projected to further increase[23], we expect that the trend towards thinner ice will continue into the future.

**Table 2 | Ice quality observations in lakes made during winter 2020/2021 (for location of the lakes see also Fig. 1a)**

| Lake Name | Country | Latitude/Longitude |
|---|---|---|
| Wilcox Lake | Canada | 43.95/−79.44 |
| Bagot Long Lake | Canada | 45.14/−76.39 |
| Harp Lake | Canada | 45.38/−79.14 |
| Lake O'Hara | Canada | 51.36/−116.33 |
| Lake Võrtsjärv | Estonia | 58.21/26.10 |
| Lake Saadjärv | Estonia | 58.53/26.65 |
| Lake Peipsi | Estonia/Russia | 58.80/27.00 |
| Lake Oulujärvi | Finland | 64.32/27.72 |
| Lake Dagow | Germany | 53.06/12.59 |
| Lake Fuchskuhle | Germany | 53.06/12.59 |
| Lake Tovel | Italy | 46.26/10.95 |
| Lake Alauksts | Latvia | 57.09/25.76 |
| Mozhaysk Reservoir | Russia | 55.58/35.86 |
| Lake Vedlozero | Russia | 61.34/32.49 |
| Lake Kroshnozero | Russia | 61.42/33.04 |
| Lake Vendyurskoe | Russia | 62.10/33.10 |
| Lake Kuropachie | Russia | 67.56/32.42 |
| Lake Imandra | Russia | 67.60/33.00 |
| Lake Big Vudyavr | Russia | 67.63/33.69 |
| Lake Small Vudyavr | Russia | 67.69/33.63 |
| Lake Sopchyavr | Russia | 67.90/32.79 |
| Lake Erken | Sweden | 59.84/18.63 |
| Inre Harrsjön | Sweden | 68.21/19.25 |
| Lake Nero | Switzerland | 46.45/08.54 |
| Falling Creek Reservoir | USA | 37.30/−79.84 |
| Long Pond | USA | 44.53/−69.84 |
| Lake Champlain | USA | 44.56/−73.24 |
| Crystal Lake | USA | 46.00/−89.61 |
| Sparkling Lake | USA | 46.01/−89.70 |
| Hobart Lake | USA | 46.92/−98.14 |
| Lake Itasca | USA | 47.23/−95.20 |

These observational data comprised the dataset named IceBlitz.

The ice on IceBlitz lakes commonly consisted of both a black and a white ice layer where the black ice layer was the dominant ice layer when all 167 measurements were taken into consideration (mean ± one standard deviation: 70 ± 28%). There was, however, a clear seasonal trend in the ice layer composition with a substantial increase in the thickness and the proportion of white ice towards late winter and spring (Fig. 2a, b). From being mainly absent in December 2020, the median white ice thickness across the Northern Hemisphere increased to 3 cm (13% of the total ice thickness) in January 2021, 10 cm (33% of the total ice thickness) in February 2021, and finally 23 cm (52% of the total ice thickness) in March 2021 (Fig. 2a, b). Thus, white ice became the dominant ice type for the 31 seasonally frozen lakes located across the Northern Hemisphere during the time before ice-off. An increase in the thickness and proportion of white ice from the beginning towards the end of ice cover was observed for all lakes with available seasonal data (see Fig. 2c for an example) except for Hobart Lake, USA, where the white ice layer slightly decreased from 9 to 4–8 cm from January to February 2021. The range of spatial variation in the accumulation of white ice on a lake during one sampling occasion remained below 8 cm even when 12 different sites on a lake were visited. Only once the in-lake difference in the white ice layer reached 21 cm on Lake

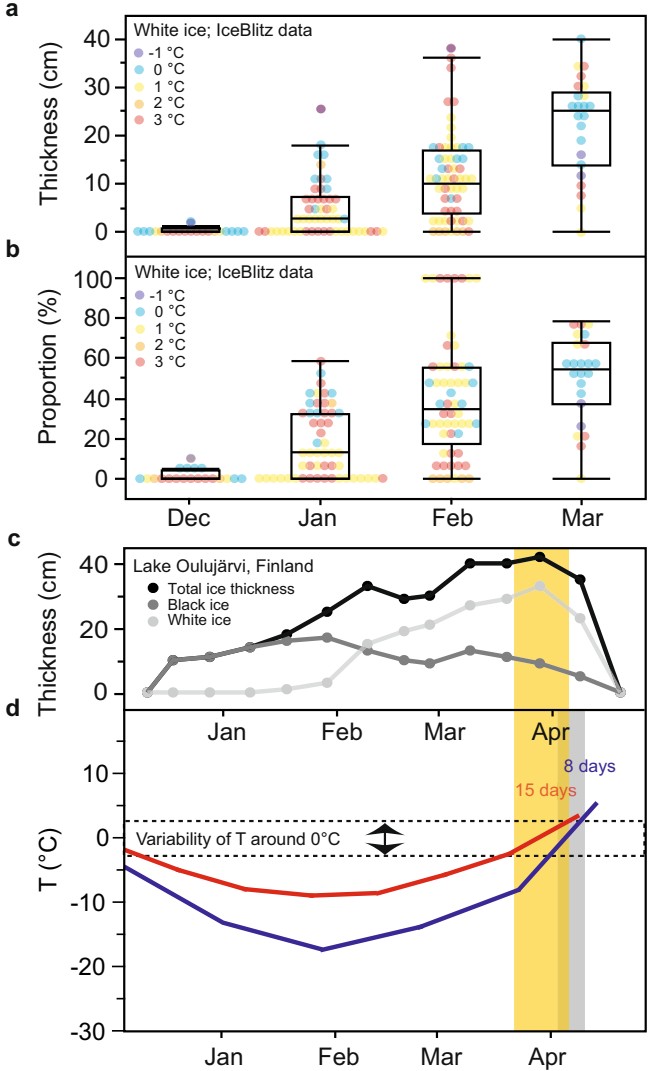

**Fig. 2 | Seasonal accumulation of white ice in lakes. a**, **b** Boxplots showing the seasonal development of the thickness of white ice and the percentage of white ice observed in 31 Northern Hemisphere lakes during the IceBlitz sampling campaign in 2020/2021. Boxplots depict the minimum, first quartile, median, third quartile, and maximum. April values are not shown because too few measurements were available from that month. Colors represent the lake site-specific mean air temperature anomaly during December through March in 2020/2021 relative to the base period 1951–1980. Except for the lakes located on the Kola Peninsula (Northwest Europe), all lakes experienced warmer than normal winter air temperatures during the IceBlitz campaign. Eight lakes even had 3 °C warmer air temperatures compared to 1951–1980. **c** Seasonal development of total ice thickness and the thickness of black and white ice in Lake Oulujärvi, Finland during 2020/2021. The orange shaded area marks the time period when air temperatures varied around the freezing point, which is relevant for the formation of white ice. **d** Simplified, typical winter air temperature (T) curves representing a cold (blue line) and a warm (red line) winter (data taken from Weyhenmeyer et al.[26]). The number of days when air temperatures vary around the freezing point increases when the seasonal cycle of winter air temperatures falling below 0 °C flattens during a warm year[26]. In our conceptual figure, the number of days when air temperatures vary around the freezing point corresponds to -15 days during a warm winter (marked in orange) compared to -8 days during a cold winter (marked in blue).

Vendyurskoe. Thus, the seasonal variation in the thickness of the white ice layer commonly exceeded the observed in-lake variation assessed during one sampling event.

## White ice formation

White ice is commonly formed when snow accumulates on ice, melts, and refreezes[24]. White ice also forms when rain falls on the snow layer to form slush, which subsequently can freeze and turn into white ice or when the snow load is sufficient to force lake water to the ice surface through cracks in the ice matrix[9,25]. Melting and refreezing of ice, slush and snow as well as rainfall are conditions that typically occur when air temperatures vary around the freezing point, i.e., usually during the time before ice-off. We attribute the ice layer composition change towards spring to changes in the seasonality of winter air temperatures. When winters become warmer, the seasonal cycle of air temperatures below the freezing point flattens, similar to the flattening that occurs towards warmer geographical regions[26]. A decrease in the winter air temperature amplitude implies that the number of days with air temperatures varying around the freezing point increases (Fig. 2d), prolonging the formation of white ice and increasing its proportion. This lake ice quality response to warmer winter air temperatures is increasingly pronounced towards warmer geographical regions[26]. Under continued global warming, lakes located in those warmer geographical regions are most likely increasingly exposed to air temperatures varying around the freezing point, resulting in an increased prevalence of white ice, in particular during the critical time before ice-off.

## White ice and ice stability

An increase in the proportion of white ice can jeopardize the use of seasonally ice-covered lakes for subsistence, recreation, transportation and other purposes.[27,28] It has already been noted that most fatal winter drownings occur just prior to ice-off[29], coinciding with the time when ice layers on lakes commonly become dominated by white ice. Gold[15] has developed a method to estimate the bearing strength of ice depending on the total thickness and ice quality:

$$P = A \cdot H^2 \qquad (1)$$

where $P$ is the allowable load in kg, $A$ is the bearing strength that varies between 3.5 and 17.5 kg cm$^{-2}$ depending on ice quality (17.5 kg cm$^{-2}$ for cold, black ice conditions of high quality, i.e., no snow ice, no cracks, no bubbles etc. and 3.5 kg cm$^{-2}$ for low quality ice based on the conclusions from Gold[15]), and $H$ is the total ice thickness in cm. Although Eq. (1) with $A = 3.5$ kg cm$^{-2}$ is used for general ice safety guidelines[30], Gold[15] described some cases in which the predicted load was not supported by the ice layer. White ice conditions at air temperatures varying around the freezing point (0 °C) might be responsible for ice collapse despite a sufficient theoretical total ice thickness. Laboratory experiments have shown a reduction of the flexural strength of ice (which is proportional to the bearing strength $A$) by 51% for white ice compared to black ice at a temperature of −0.5 °C[16]. Based on these results, we suggest a modification of Gold's equation during days when air temperatures vary around the freezing point:

$$P = \frac{A \cdot H^2}{2} \cdot \left(1 + \frac{100 - \% white\, ice}{100}\right) \qquad (2)$$

where $P$ is the allowable load in kg, $A$ corresponds to 3.5 kg cm$^{-2}$ (see also Eq. (1)), $H$ is the total ice thickness in cm and % white ice is the proportion of white ice in the ice layer. Further adjustments of Eq. (2) might be needed when air temperatures stay above 0 °C for 24 h or more because under those conditions ice usually loses additional strength[15,31]. It also important to keep in mind that Eq. (2) will not apply to ice conditions just before ice-off when the entire ice matrix begins to deteriorate.

Applying Eq. (2), an individual weighing 100 kg might safely walk on ice if there is a white ice layer of at least 8 cm thickness (Fig. 3a).

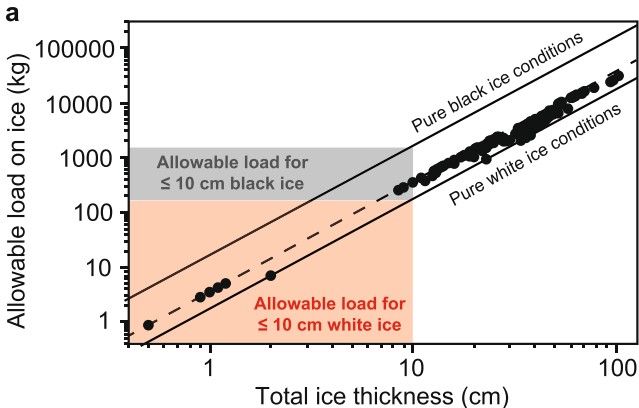

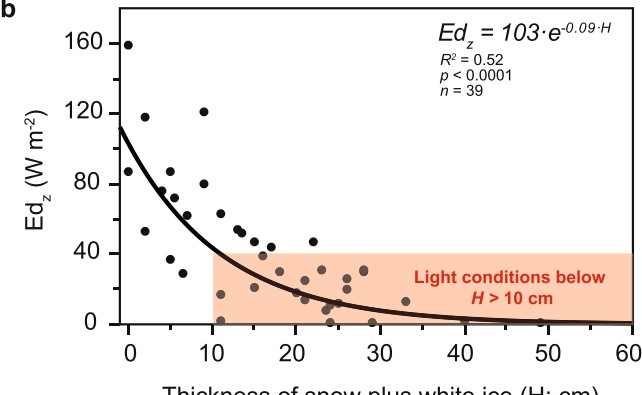

**Fig. 3 | Lake ice conditions that are critical for ice stability and for the transmittance of photosynthetically active radiation. a** Variation in the estimated allowable load on ice depending on total ice thickness and ice quality. Pure black ice conditions were modeled using Eq. (1) with $A = 17.5$ kg cm$^{-2}$ and pure white ice conditions using Eq. (2) with $A = 3.5$ kg cm$^{-2}$. The dashed line represents estimates of the allowable load using Eq. (1) with $A = 3.5$ kg cm$^{-2}$, which is commonly used for ice safety guidelines. Black dots show estimates for the IceBlitz dataset using Eq. (2) with $A = 3.5$ kg cm$^{-2}$. The red and gray shaded areas mark the allowable load for an ice thickness of 10 cm and less under pure white ice (red) and pure black ice (gray) conditions. **b** Daily mean (8.00 a.m. to 8.00 p.m.) under-ice irradiance ($Ed_z$) in Watts (W) m$^{-2}$ in Lake Vendyurskoe during spring just before ice-off in relation to the thickness of snow and white ice ($H$) on the lake. Data were taken from Zdorovennova et al.[36], measured during 1997–2020. Shown is the exponential decline of $Ed_z$ with increasing $H$ (black line). The red shaded area marks the light availability below a 10 cm thick snow and white ice layer on a lake.

Thus, the recommended 10 cm of total ice thickness on lakes, which most webpages give as an ice safety guideline e.g.,[32], seems generally suitable for one single person to safely walk on ice, as this ice thickness allows for a load between 175 and 1753 kg depending on ice quality (Fig. 3a). It is, however, important to keep in mind that usually more than one person walks on the ice at a given time. For example, the weight of a group of three people often exceeds 175 kg, which is the allowable load for 10 cm of pure white ice conditions (Fig. 3a). Consequently, groups of people are at higher risk to fall through ice under complete white ice conditions, despite an ice thickness of 10 cm or more. We observed such complete white ice conditions on eight occasions in three different countries during the IceBlitz campaign, i.e., in Canada (Lake Wilcox), Switzerland (Lake Nero) and Sweden (Lake Erken). All 100% white ice conditions occurred in February 2021, when the total ice thickness in the three lakes varied between 2 and 38 cm, with a median of 12.5 cm. In central and northern Sweden, February is usually the month when people can safely walk on ice, and rarely are any fatal winter drownings reported. In February 2021, however, ten people died by falling through ice which represented the

highest winter drowning death rate in Sweden during the month of February since the records began in 2000[33]. The ice situation in February 2021 demonstrates that behavioral adaptations to a warmer world are needed. Ice conditions that traditionally have been safe during winters of the past will become unsafe in the future. One strategy to make people more aware of the increasing risk of unstable ice conditions in a warmer world might be to spread information on revised ice safety guidelines highlighted above. To keep the update of the ice safety guidelines as simple as possible a rule of a thumb could be to double the presently used ice thickness guideline.

### Light transmittance through white ice and ecological effects

White ice conditions do not only affect the stability of the ice cover but also the light regime underneath. White ice is known to have a severalfold higher reflectance than black ice, thus only small amounts of photosynthetically active radiation penetrate through white ice[34]. The transmittance of light through a combined black and white ice/snow layer on lakes has previously been described by a two-layer model[34]:

$$\tau(C) = \frac{\tau(A) \cdot \tau(B)}{1 - \rho(A) \cdot \rho(B)} \qquad (3)$$

where $\tau(C)$ is the transmittance of light through the combined black and white ice/snow layer, $\tau(A)$ is the transmittance of light through the upper white ice/snow layer, $\tau(B)$ is the transmittance of light through the lower black ice layer, $\rho(A)$ is the reflectance of the upper white ice/snow layer and $\rho(B)$ is the reflectance of the lower black ice layer. $\tau(C)$ approaches zero when snow covers the ice, as snow usually has a reflectance of close to one[35]. $\tau(C)$ also declines when the upper white ice layer becomes thicker[36]. Based on the data from Lake Vendyurskoe (1997–2020)[36] we obtained the following model for underneath ice light conditions:

$$Ed_z = a \cdot e^{-\varepsilon \cdot H} \qquad (4)$$

where $Ed_z$ is the 12-hour day time mean (8.00 a.m. to 8.00 p.m.) irradiance underneath ice in W m$^{-2}$, $a$ is a solar radiation dependent constant (in our example corresponding to a value of 103; see Fig. 2b), $\varepsilon$ is the extinction of light with 0.09 W m$^{-2}$ cm$^{-1}$ and $H$ is the thickness of the sum of snow and white ice on a lake in cm. When we included the thickness of black ice in $H$, the prediction of $Ed_z$ did not improve ($R^2 = 0.45$, $p < 0.0001$, $n = 39$ compared to $R^2 = 0.52$, $p < 0.0001$, $n = 39$ when only white ice and snow was considered). We attribute the lack of model improvement to the fact that the thickness of black ice was unrelated to $Ed_z$ in Lake Vendyurskoe ($p > 0.05$). These results indicate that $Ed_z$ is mainly driven by the thickness of snow and white ice, where each cm of snow and white ice reduces the under-ice irradiance by approximately 9%. Further, by analyzing the inter-annual variability in $Ed_z$ in Lake Vendyurskoe in spring, we found no trend over time in $Ed_z$ despite a significant decrease in total ice thickness during this period (linear time series model: $p > 0.05$ and $p = 0.007$, respectively). Thus, under-ice light conditions did not improve despite thinner ice in spring. We suggest that these under-ice light conditions were driven by the thickness of white ice which did not show a significant decrease in spring from 1997 to 2020 (linear time series model: $p > 0.05$).

Low light conditions in spring caused by a white ice layer and/or snow on ice are critical for the development of primary producers and consumers as their growth and reproduction is dependent on light and convection underneath ice[37,38]. Most vulnerable to white ice conditions and snow on ice are photoautotrophs in lakes[39], in particular non-motile photoautotrophs that are dependent on radiatively-driven convection to reach sufficient light (e.g., diatoms)[40,41]. Motile taxa can have an advantage during low light conditions under-ice[13]. Mixotrophy

is another strategy to adapt to low-light under-ice conditions[42]. Further up in the food web zooplankton has developed survival strategies to low-light under-ice conditions by feeding on mixotrophic microorganisms and allochthonous organic material, but for growth and reproduction, a diet containing nutrient-rich photoautotrophs is still essential[43]. Thus, despite a variety of survival strategies, snow and white ice induced changes to the photoautotroph community cascade through the food web, with substantial consequences for microbial, zooplankton, and fish populations[44,45].

### Relevance and future perspective
Our sampling campaign focused on spatial and seasonal patterns. The most obvious and relevant result was the consistent buildup of white ice in spring until the time just before ice-off which we attribute to the increased time window when air temperatures vary around the freezing point. Under expected future global warming, many lakes in the world will become exposed to these prolonged freeze-thaw cycles. According to our study, such lakes will experience a shift in the ice layer composition towards white ice conditions. We conclude that white ice conditions in lakes need far more consideration than previously given. We suggest the use of Eq. (2) to re-examine ice safety guidelines to account for ice instability during white ice conditions. We also recommend considering the thickness of white ice as an important regulator of physical, chemical, and biological processes in lakes.

## Methods
### IceBlitz sampling campaign
We performed a global sampling campaign, named IceBlitz, within the Global Lake Ecological Observatory Network (GLEON, https://gleon.org) from December 2020 to April 2021. Participants of the campaign were asked to conduct at least one sampling visit at an ice-covered lake and measure (preferably at several locations on the same lake) total ice thickness, the thickness of black and white ice, the thickness of snow and/or slush layers on top of the ice layer, water temperature in the drilling hole and air temperature 1.5 m above the drilling hole using a standardized protocol (Supplementary Information). The sampling took place in 31 lakes across the Northern Hemisphere (Fig. 1a, Table 2). Altogether 167 ice quality measurements were made. All data are open access (see data availability). A detailed data description for Hobart Lake is available as a USGS data release[46]. Two out of the 31 lakes had additional ice quality data available from previous years, i.e., Mozhaysk Reservoir (15 years of ice quality data during 1971–2021) and Lake Vendyurskoe (ice quality data during spring before ice-off during 1997–2021). We used these data to evaluate in how far ice conditions during previous years differed from our sampling winter. Additionally, we collected literature data on ice quality, which comprised 117 ice observations from 18 lakes across the Northern Hemisphere (Fig. 1a, Table 1).

### Meteorological data
In addition to ice quality data, we analyzed air temperature data, i.e., Northern Hemisphere-mean monthly, seasonal, and annual means from 1880 to present, downloaded from GISTEMP Team, 2022: GISS Surface Temperature Analysis (GISTEMP), version 4. NASA Goddard Institute for Space Studies[47,48]. The data are given as anomalies over the base period 1951 to 1980 (Fig. 1b). Data from the GISTEMP team were also used to create a map showing the December through March air temperature anomaly across the Northern Hemisphere in 2020/2021. Maps were created on 2022-02-18 via https://psl.noaa.gov/data/gridded/data.gistemp.html, choosing variable: air temperature and statistics: Monthly Anomaly: 250 km smoothed. On the maps, we located each lake from the IceBlitz campaign and allocated a December through March temperature anomaly in 2020/2021 to each lake (Fig. 2a, b).

### Statistics
All statistical analyses, except of the time series analyses, were performed in JMP, version 14.2.0., SAS Institute Inc. For the smoothing splines a lambda of 0.05 was chosen which commonly is used to estimate a functional relationship between a predictor and a response variable. Boxplots are outlier boxplots, also known as box-and-whisker plots, where the bottom and top of the box show the 25th and 75th quantiles and the lines that extend from the box are whiskers that represent 1.5 times the interquartile range from the top and bottom of the box. Values that fall above or below the end of the whiskers are plotted as dots. Time series analyses were performed in R 4.0.2 (R Core Team, 2020) using linear models and checking for temporal autocorrelation, i.e., lm(formula = ice-time). Time series analyses were performed on April measurements for Lake Vendyurskoe. Data from Mozhaysk Reservoir were not used for time series analyses because ice quality was only measured during 15 out of 51 years from 1971 to 2021.

## Data availability
The ice quality data generated in this study have been deposited in the Swedish DIVA database and are available at http://urn.kb.se/resolve?urn=urn:nbn:se:uu:diva-468290. All data are open access.

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

## Acknowledgements

The work was carried out within the Global Lake Ecological Observatory Network (GLEON). G.A.W., H.R. and E.J. received financial support for this study from the Swedish Research Council (Grant no. 2020-03222) and Swedish Research Council for Environment, Agricultural Sciences and Spatial Planning (FORMAS; Grant no. 2020-01091). A.L. was funded

by the Estonian Research Council (grants PSG32, PRG709). G.Z., R.Z. and N.P. conducted sampling within the framework of the project FMEN-2021-0019 of the state assignment of the Russian Federation. G.Z., R.Z. and I.F. received financial support for field measurements from Russian FBR (Grant no. 18-05-60291). J.M.F. received funding from the National Science Foundation (NSF-DEB 1754181) and thanks the Parks Canada Agency for permission to conduct research in the national mountain parks of Canada. Sampling of Falling Creek Reservoir was supported by NSF DEB-1753639. For practical and technical support we would like to thank Adrienne Breef-Pilz, Andrey Mitrokhov, Anni Karppinen, Benton Fry, Maria Chernyshova, Mark H. Olson, Matthew Futia, Mattia Domenici, Michael Sachtleben, Monika Degebrodt, Olivia Johnson, Stefano Rioggi, Solvig Pinnow, Thomas Buhl and Uta Mallock. All ice quality observation data from Bagot Long Lake were kindly measured and provided by William Allison. Any use of trade, firm, or product names is for descriptive purposes only and does not imply endorsement by the U.S. Government.

## Author contributions

G.A.W. initiated and led the study, completed the analysis and wrote the manuscript with input from all co-authors, i.e., U.O., H.R., E.J., J.J., G.Z., S.B., B.D.B., C.C.C., J.P.D., H.D., O.E., I.F., J.M.F., L.G., H.P.G., K.K., L.B.K., A.L., F.L., J.M., N.P., M.P., M.P., J.A.R., S.S., D.W. and R.Z. U.O. contributed to data analyses and performed the time series analyses. H.R. prepared the literature dataset. E.J. gave substantial input to under-ice ecology, J.J. to ice stability and G.Z. to ice formation and decay. S.B., B.D.B., C.C.C., H.D., O.E., J.M.F., L.G., H.P.G., J.J., K.K., L.B.K., A.L., F.L., J.M., M.P., U.O., M.P., J.A.R., S.S. and D.W. participated in the IceBlitz campaign, i.e., they gave input to the design of ice protocols, sampled, prepared and contributed with ice quality data. G.Z., I.F., N.P. and R.Z. did not officially participate in the IceBlitz campaign but measured ice quality in a variety of Russian lakes during the winter 2020/2021.

## Funding

## Competing interests

The authors declare no competing interests.
