## [Peer Review File · Nature Communications]

Towards critical white ice conditions in lakes under global warmingReviewers' Comments:

Reviewer #1:

Remarks to the Author:

Weyhenmeyer et al. presented a valuable dataset of lake ice quality across the entire Northern Hemisphere, and their main objective is to evaluate the changes in lake ice conditions as compared to historical periods. Although the reviewer appreciates the substantial efforts of data collection, the paper seems to have two major limitations and many other issues that prevent its publication in NC. Major limitations

1. The comparison between the IceBlitz and literature data does not make sense to me. For example, the two datasets from different lakes (Table 1 and Table 2); different lakes show marked disparities in lake ice features, due to their distinct locations, sizes, depths, morphological and hydrothermal conditions, and many other factors. Moreover, the sampling time for the two datasets may also vary significantly (I did not check the references for the sampling period of historical data), causing the mismatches in the associated lake ice qualities.
2. For such a short letter, I would suggest simplifying the discussion of the relationship between ice thickness and ice strength. The current manuscript used $\sim 1/3$ of the space to present this issue, which was mostly based on previous studies. Moreover, the modification of Gold's equation was based purely on speculations without any validations.

Other comments:

1. Line 45. Cite the standardized protocol.
2. Line 46. From how many lakes?
3. Line 50. I appreciate the objective, but the comparison is not valid.
4. Fig. 1 Caption. Can you explain a bit "smoothing splines using a lambda of 0.05"? Fig. 1b. The trends are basically the same for the five panels.
5. Line 53-55. What are the sampling seasons for the IceBlitz and historical data? Or annual mean? The statistics were from how many lakes? How about the standard deviations?
6. Line 57. Not true, please see the major comment.
7. Line 60-63. I expect a similar analysis for other lakes
8. Line 67-90. Can be moved into the method section.
9. Line 92-113. Suggest simply and move into the discussion section.
10. Line 111. Not clear the to "revise" the safety guidelines for ice thickness or winter month?
11. Line 127. Not necessarily true.
12. Line 128-144. Should be moved into the method section.
13. Line 144-148. Maybe I have missed something, but these statements are not well supported by previous calculations.

Reviewer #2:

Remarks to the Author:

The manuscript by Gesa Weyhenmeyer and colleagues uses data from a field campaign across the Northern Hemisphere to investigate white ice during the 2020/2021 ice season. The authors find that the proportion of white ice was high and it was related to a warmer than usual winter. This has significant safety implications in relation to future climate change, in particular the influence of rising temperatures and its impact on lake ice.

I find the paper well-written and clearly argued. I have no concerns on the underlying science, but there are a couple of areas where I think the manuscript could be improved. I briefly summarise these below, but they are elaborated upon more in the attached pdf, along with some other small comments throughout the text that could improve clarity.

- I think the abstract needs a little work to highlight the broader results before then referring to the

specifics of the warm winter. The abstract is also missing some geographical information mentioning where the study sites are in the context of the results. I also feel that the take home message in relation to future change and white ice safety is not clearly made and in its current format undersells the significance of the paper.

- The discussions comparing the authors' results with those from the literature misses the temporal component of the literature data. Including this in the narrative would make the arguments by the authors more robust. I see no reason to question their conclusions, but I think this would be helpful.

In conclusion, I like this paper and believe it should be published after some minor revisions. Congratulations to Professor Weyhenmeyer and colleagues for this interesting work. If any of my comments are not clear then the authors are very welcome to contact me.

Andrew Newton - Queen's University Belfast

Point-by-point response to the reviewers' comments

Comments	Reply
Reviewer 1 Weyhenmeyer et al. presented a valuable dataset of lake ice quality across the entire Northern Hemisphere, and their main objective is to evaluate the changes in lake ice conditions as compared to historical periods. Although the reviewer appreciates the substantial efforts of data collection, the paper seems to have two major limitations and many other issues that prevent its publication in NC. Major limitations 1. The comparison between the IceBlitz and literature data does not make sense to me. For example, the two datasets from different lakes (Table 1 and Table 2); different lakes show marked disparities in lake ice features, due to their distinct locations, sizes, depths, morphological and hydrothermal conditions, and many other factors. Moreover, the sampling time for the two datasets may also vary significantly (I did not check the references for the sampling period of historical data), causing the mismatches in the associated lake ice qualities. 2. For such a short letter, I would suggest simplifying the discussion of the relationship between ice thickness and ice strength. The current manuscript used	We agree that a comparison with literature data has limitations, and therefore, we made major revisions accordingly (specified below). We, however, would like to emphasize that the comparison with literature data was not a main objective of the manuscript. In fact, the literature data had only been used for a complementary comparison outlined in one short paragraph. None of the major results has a connection to the literature data. So, we rearranged the entire manuscript and removed the sentence where we previously wrote that our intention was to compare data to previous literature. Instead of comparing our results with literature data, we now performed a variety of additional data analyses, focusing on our own data set. We agree that the comparison has limitations and because the comparison has no influence on the main results, we removed the comparison from our objectives. We now discuss literature data only to provide context to our own results and only once. We also added the limitations whenever data from different lakes and time periods are compared. We agree that the discussion on ice safety issues is rather long. However, for scientific papers it is not uncommon that the text length of a discussion exceeds the text length on the actual results by a substantial factor. Our results concern the thickness and proportion of white ice conditions found in the Northern Hemisphere during a very warm winter. Everything else is

~1/3 of the space to present this issue, which was mostly based on previous studies. Moreover, the modification of Gold's equation was based purely on speculations without any validations.	discussion of the results. To further perform laboratory and field experiments is beyond the scope of the study, but could of course be pursued in future studies. We assume that the reviewer might have been a bit confused by the structure of our manuscript. In our previously submitted version, it was not very clear which sentences belonged to results and which to the discussion. To avoid any confusion, we now rearranged the entire manuscript where we first describe all results before we begin with our main discussion. The changes include a clearer reference to published laboratory experiments on the flexural strength of white ice versus black ice (ref. 16 in the main text).
Line 45. Cite the standardized protocol.	Done. We added the sampling description and the protocol as Supplementary Information.
Line 46. From how many lakes?	This information has now been added.
Line 50. I appreciate the objective, but the comparison is not valid.	We agree and consequently removed the comparison from our objectives.
Fig. 1 Caption. Can you explain a bit "smoothing splines using a lambda of 0.05"? Fig. 1b. The trends are basically the same for the five panels.	Smoothing splines are a powerful approach for estimating functional relationships between a predictor X and a response Y, but they should not be mixed up with an actual temporal trend in Kelvin per time unit. A smoothing spline only "smoothes" data variability similar to a moving average. A large value for lambda makes the fit of the data stiff (less curved) while a small value for lambda gives the error term of the spline model more weight and the fit becomes more flexible and curved. The smoothing splines look so similar between different months because of the similarity of data variability between months. Because figure captions should be kept short, we decided not to add any further information in the caption but instead added a sentence in the method part, justifying the choice of lambda.
Line 53-55. What are the sampling seasons for the IceBlitz and historical data? Or annual mean? The statistics were from how many lakes? How about the standard deviations?	We agree that some information was lacking and added additional statistical results to the manuscript.
Line 57. Not ture, please see the major comment.	See reply above. We rearranged the entire paragraph.

Line 60-63. I expect a similar analysis for other lakes	Since a study of total ice thickness was not the main purpose of the study, we decided to rely on already published results on temporal changes in total ice thickness, in particular because our analysis would also only rely on already published data.
Line 67-90. Can be moved into the method section.	Here, we do not fully agree because these sentences concern a discussion of our results. To move it to the method section would give the impression that it was our main purpose to evaluate the strength of ice. To avoid any misunderstanding, we rearranged the entire manuscript.
Line 92-113. Suggest simply and move into the discussion section.	We previously did not clearly differentiate between methods, results, and discussion, but now we followed the classical division into different sections a bit better to avoid any misunderstandings.
Line 111. Not clear the to “revise” the safety guidelines for ice thickness or winter month?	We agree that this section may not have been totally clear and clarified it in the revision.
Line 127. Not necessarily true.	We removed this sentence.
Line 128-144. Should be moved into the method section.	See our reply above (line 67-90). A rearrangement of the entire manuscript should help to understand why this part has not been moved to the method section.
Line 144-148. Maybe I have missed something, but these statements are not well supported by previous calculations.	We agree that we should give more concrete statistical results, which we now added.
Reviewer 2 I find the paper well-written and clearly argued. I have no concerns on the underlying science, but there are a couple of areas where I think the manuscript could be improved. I briefly summarise these below, but they are elaborated upon more in the attached pdf, along with some other small comments throughout the text that could improve	Thank you very much. Please see our responses to the specific comments below.

clarity.

- I think the abstract needs a little work to highlight the broader results before then referring to the specifics of the warm winter. The abstract is also missing some geographical information mentioning where the study sites are in the context of the results. I also feel that the take home message in relation to future change and white ice safety is not clearly made and in its current format undersells the significance of the paper.

- The discussions comparing the authors' results with those from the literature misses the temporal component of the literature data. Including this in the narrative would make the arguments by the authors more robust. I see no reason to question their conclusions, but I think this would be helpful.

In conclusion, I like this paper and believe it should be published after some minor revisions. Congratulations to Professor Weyhenmeyer and colleagues for this interesting work. If any of my comments are not clear then the authors are very welcome to contact me.

Comments given directly in the manuscript.

We revised the abstract. We, however, decided not to oversell our results. We have no evidence of what will happen in future. Because we rely mainly on spatial and seasonal patterns, we believe that we should keep the implications at the current level, even though mechanisms point into the direction of an increasing occurrence of life-threatening white ice conditions in future.

This point is similar to what reviewer 1 has pointed out. Please see our reply above to reviewer 1. We removed the comparison now, but instead added other statistics to confirm our main message.

Thank you very much for this favorable comment.

We considered/accepted all comments directly given in the manuscript and changed accordingly except of:
The statement that most lakes in the world are still seasonally ice covered is correct since the abundance of lakes is highest in the boreal and arctic region

“The response of lake ice cover to increasing air temperatures is nonlinear” already says that the nonlinearity refers to air temperature so that we do not really know why the reviewer asked “nonlinear to what?”

To mark three lakes on figure 1a because it would result in a longer figure caption. The three lakes can easily be looked up by readers, in particular since countries are specified.

End of reply

Reviewers' Comments:

Reviewer #1:

Remarks to the Author:

I appreciate the authors in addressing my comments, especially in rearranging the manuscript, and turning down the tune on the comparisons with previous studies. I like the current version of this paper, and recommend it for publication.

Reviewer #2:

Remarks to the Author:

This paper is a revised version and my previous comments, along with those of the other reviewer, have been largely addressed in my opinion. The main concern I had about the discussion around the temporal context of other published work has been largely removed and replaced with statistical analyses – I think this is a good call and makes for a cleaner piece of work. The paper is well written with appropriate figures. This is an interesting piece of work that I am sure will fuel some further discussion – indeed, if published, I expect to use it for an undergraduate tutorial to discuss the issues of white ice. I only have a handful of minor comments below (including some clarification on one of my previous comments). I am happy to recommend accepting this paper for publication.

Andrew Newton
Queen's University Belfast

Line 4: Change to "...load-bearing strength..."

Line 24: My previous comment on the manuscript was insufficiently clear. When I commented about "nonlinear to what?" I was looking for additional information on what you mean when you say "the response", what exactly do you mean by response? Is it a response in thickness/area/quality/temperature, maybe all. A non-expert in this specific topic might benefit from that extra clarity.

Line 77: "...reported in the literature for the same study sites" – suggested edit, though I am not quite sure whether what you mean by "the literature" is specifically referring to those sites. Perhaps clarify.

Figure 1: As a point of principle, on something as important as a study site for a paper, I do not believe it is reasonable to expect a reader to have to go elsewhere to clarify something as simple as where it is (unless the paper has 100s of study sites where it would be laborious to do so). Ultimately, while I would prefer the sites mentioned by name to be labelled, I am happy to leave it to the judgement of the authors. I previously suggested some different coloured circles and addition of a few extra words to the caption, but perhaps a better suggestion would be to provide a larger supplementary map that labels all the study sites, perhaps with numbers – you might consider adding numbered labels to Table 1 as well and that would make it very simple to cross-reference.

Point-by-point response to the reviewers' comments

Comments	Reply
Reviewer #1 (Remarks to the Author): I appreciate the authors in addressing my comments, especially in rearranging the manuscript, and turning down the tune on the comparisons with previous studies. I like the current version of this paper, and recommend it for publication. Reviewer #2 (Remarks to the Author): This paper is a revised version and my previous comments, along with those of the other reviewer, have been largely addressed in my opinion. The main concern I had about the discussion around the temporal context of other published work has been largely removed and replaced with statistical analyses – I think this is a good call and makes for a cleaner piece of work. The paper is well written with appropriate figures. This is an interesting piece of work that I am sure will fuel some further discussion – indeed, if published, I expect to use it for an undergraduate tutorial to discuss the issues of white ice. I only have a handful of minor comments below (including some clarification on one of my previous comments). I am happy to recommend accepting this paper for publication. Andrew Newton Queen's University Belfast ----- Line 4: Change to "...load-bearing strength..." Line 24: My previous comment on the manuscript was insufficiently clear. When I commented about "nonlinear to what?" I was looking for additional information on what you mean when you say "the response", what exactly do you mean by response? Is it a response in thickness/area/quality/temperature, maybe all. A non-expert in this specific topic might benefit from that extra clarity.	Thank you very much. Thank you very much. We agree. The sentence was however now removed because we had too many words in the abstract. Thank you for the clarification. Now we do understand the comment. We changed the sentence to: Increasing temperatures have an effect on the mean and amplitude of the annual air temperature cycle, which in turn causes accelerated ice cover loss rates in a warmer world⁵, which have been documented for many lakes across the

Line 77: "...reported in the literature for the same study sites" – suggested edit, though I am not quite sure whether what you mean by "the literature" is specifically referring to those sites. Perhaps clarify. Figure 1: As a point of principle, on something as important as a study site for a paper, I do not believe it is reasonable to expect a reader to have to go elsewhere to clarify something as simple as where it is (unless the paper has 100s of study sites where it would be laborious to do so). Ultimately, while I would prefer the sites mentioned by name to be labelled, I am happy to leave it to the judgement of the authors. I previously suggested some different coloured circles and addition of a few extra words to the caption, but perhaps a better suggestion would be to provide a larger supplementary map that labels all the study sites, perhaps with numbers – you might consider adding numbered labels to Table 1 as well and that would make it very simple to cross-reference.	Northern Hemisphere over the past decades^{3,6} We agree that our statement "reported in the literature" was not clear. We rephrased the sentence and added Table 1: "The observed ice thickness during the 2020/2021 IceBlitz campaign was generally lower than we found reported in the literature (Table 1). Since it is the second time that this point is raised we now added all lake names in Figure 1. It looks a bit crowded but hopefully it is doable.
--	--

End of reply